# Novel Cellular Functions of ATR for Therapeutic Targeting: Embryogenesis to Tumorigenesis

**DOI:** 10.3390/ijms241411684

**Published:** 2023-07-20

**Authors:** Himadri Biswas, Yetunde Makinwa, Yue Zou

**Affiliations:** Department of Cell and Cancer Biology, College of Medicine and Life Sciences, University of Toledo, Toledo, OH 43614, USA; himadri.biswas@utoledo.edu (H.B.); yetunde.makinwa@utoledo.edu (Y.M.)

**Keywords:** ATR, DNA damage responses, DNA damage checkpoint signaling, embryogenesis, tumorigenesis, apoptosis, cancer therapeutics, prolyl isomerization

## Abstract

The DNA damage response (DDR) is recognized as having an important role in cancer growth and treatment. ATR (ataxia telangiectasia mutated and Rad3-related) kinase, a major regulator of DDR, has shown significant therapeutic potential in cancer treatment. ATR inhibitors have shown anti-tumor effectiveness, not just as monotherapies but also in enhancing the effects of standard chemotherapy, radiation, and immunotherapy. The biological basis of ATR is examined in this review, as well as its functional significance in the development and therapy of cancer, and the justification for inhibiting this target as a therapeutic approach, including an assessment of the progress and status of previous decades’ development of effective and selective ATR inhibitors. The current applications of these inhibitors in preclinical and clinical investigations as single medicines or in combination with chemotherapy, radiation, and immunotherapy are also fully reviewed. This review concludes with some insights into the many concerns highlighted or identified with ATR inhibitors in both the preclinical and clinical contexts, as well as potential remedies proposed.

## 1. Introduction

Most malignancies have genomic instability, which can cause oncogenesis. In response to DNA damage, cells have evolved several methods to protect their genome. These include the checkpoint mechanism, which protects the genome by limiting cell cycle progression in the event of DNA damage. ATR (ATM and Rad3-related), a member of the PIKK (phosphatidyl inositol 3′ kinase-related kinases) protein family, is a crucial protein in checkpoint responses. ATR is activated by both DNA-damaging agents and halted replication forks. The formation of DNA damage-induced elongation replication protein A (RPA)-coated single-stranded DNA (RPA-ssDNA) during replication stressors and during DNA repair is a common theme that leads to ATR activation [1,2]. ATR interacts with a nuclear partner, ATRIP, to form the ATR–ATRIP complex, which is recruited to the sites of DNA damage by RPA-ssDNA [3,4]. The recruitment of the ATR–ATRIP complex (ATR–ATRIP) to RPA-ssDNA, on the other hand, is insufficient for ATR activation. ATR is autophosphorylated at its T1989 residue [5] after being recruited to DNA damage sites. TopBP1 (topoisomerase 2 binding protein 1) binds at this phosphorylated residue and in turn increases ATR kinase activity [6]. At dsDNA-ssDNA junctions, TopBP1 appears to interact with the Rad9–Rad1–Hus1 (9-1-1) complex [7,8,9]. TopBP1 then directly stimulates the ATR–ATRIP kinase [7,8,10]. The RPA binding protein, Ewing’s tumor-associated antigen 1 (ETAA1), interacts with RPA and functions at stalled replication forks [11,12]. ETAA1, like TopBP1, directly activates ATR–ATRIP [11,12]. In the case of severe damage, activated ATR also stimulates the activities of several key downstream proteins, e.g., p53 and other checkpoint kinases such as Chk1, resulting in an S phase cell cycle interruption to repair DNA damage or apoptosis [2,5,7,8,12,13,14,15,16,17] (Figure 1). The substrates and effectors of ATR and Chk1 have been identified to be a growing collection of DNA replication, repair, or cell cycle proteins. In particular, Chk1 phosphorylation of Cdc25 phosphatases is relevant to DNA damage inducing cancer cell cycle arrest [18]. To correct DNA replication in cells exposed to replication stress, the phosphorylation by ATR of WRN, SMARCAL1, and FANCI is essential [19,20,21]. ATR is also involved in the regulation of numerous DNA repair mechanisms, including homologous recombination, inter-strand crosslink repair, and nucleotide excision repair [22,23,24,25,26]. In particular, the ATR–Chk1 pathway not only responds to external DNA injury and stress, but it also responds to endogenous replication challenges such as those resulting from oncogenic events [27]. It has been shown to be a promising target for therapeutical intervention due to these characteristics of the ATR–Chk1 pathway. ATR serves critical roles in cellular processes in the cytoplasm, in addition to its well-established roles in DNA damage checkpoint signaling and DDR in the nucleus.

## 2. Role of ATR–Chk1 Pathway in Oncogenesis

In mouse models, homozygous ATR deletion is embryonically lethal, and in mouse embryonic fibroblasts that are acutely genetically inactivated by ATR, one or two rounds of DNA replication are required to leave the cell cycle permanently [4,28]. In contrast, a mouse model expressing kinase-dead ATR (ATR^+/KD^) but not ATR loss (ATR^−/−^) exhibited ssDNA-dependent defects at the non-homologous region of X–Y chromosomes during male meiosis, resulting in sterility, and at telomeres, rDNA, and fragile sites during mitosis, resulting in lymphocytopenia [29]. Prompt ATR exchange at DNA damage sites requires ATR kinase activity, resulting in enhancement of Chk1 phosphorylation. ATR-KD traps a fraction of ATR and RPA to chromatin, where ATM/DNA-PKcs hyperphosphorylate RPA and prevent subsequent repair [29]. Because of the replication stress resulting from spontaneously defective DNA, and particularly hard to replicate areas in the genome such as fragile regions, an ATR pathway is essential. Furthermore, genetic inactivation of ATR in adult mice causes premature aging, abnormalities in tissue homeostasis, and progenitor cell depletion in high proliferative regions [30,31]. However, it is essential to point out that life does not have to be impaired by interruption of the ATR signal. ATR levels were greatly decreased in human Seckel Syndrome, as well as in the mouse model of this syndrome by a homozygous mutation that creates mRNA splicing defects [32]. Similarly, these mice were remarkably normal in a hypomorphic model with reduced ATR levels to 10% of normal and showed no defects even in highly proliferative tissues [33].

ATR, on the other hand, is more critical in many tumor cells than it is in normal ones. First, oncoproteins such as the Ras isoforms, Myc, and Cyclin E induce dysregulated signaling that disrupts normal cell cycle regulation and causes replication stress [34]. The ATR pathway plays an important role in the survival of these tumor cells, and several studies have shown that blocking this pathway has a preferentially lethal effect on cells with high levels of oncogene-induced replication stress [33,35,36,37,38,39,40,41]. Second, deficiency in ATM makes the cells more sensitive to ATR inhibition in cell culture and in animal models [42,43,44], a finding that has led to clinical trials of an ATR inhibitor in tumors characterized by ATM deficiency or ATM mutations. Third, in cell line models, loss of specific DNA repair proteins (e.g., XRCC1, ERCC1) causes tumor cell lines more sensitive to ATR inhibition [45,46,47,48]; however, these findings are yet to be implemented in animal models. Fourth, hypoxic cells, which are typically resistant to chemo- and radiation therapy [49], are sensitive to ATR inhibition [50,51,52], likely because hypoxia causes replication stress [53]. Fifth, tumor cells that rely on the alternative lengthening of telomeres (ALT) pathway are also more sensitive to ATR disruption due to ATR’s role in the homologous recombination reactions that maintain these telomeres [54]. Thus, taken together, these observations indicate that multiple events that drive tumorigenesis may create a synthetic lethality for ATR inhibition such as the finding that PARP inhibition is synthetically lethal in BRCA1/2-deficient tumors [55].

In particular, the therapeutic benefit of ATR inhibitors exceeds malignancies with these characteristics. ATR helps cells survive by preventing cell cycle advancement, maintaining halted replication forks, and promoting DNA repair, including homologous recombination. Even in cells where inhibiting/deactivating ATR has minimal or no cytotoxic effects, these ATR inhibitors are highly sensitive to or work in conjunction with all genotoxic drugs studied. The possibility of similar sensitization or constructive collaboration in normal cells is a key worry with any drug that boosts the cytotoxic activity of a genotoxin. The findings thus far have alleviated that fear. Multiple investigations have indicated that ATR inhibitors synergize with genotoxins more efficiently in cells with p53 abnormalities than in p53-proficient cells [37,42,44,48,56] as a result of the loss of the G1 checkpoint and/or higher replication stress caused by relaxed S-phase entrance [34]. This suggests that in cancers with p53 pathway abnormalities, ATR inhibitors may be able to treat them at the clinic.

The replication stress in cancer cells is much higher than in normal cells due to activated oncogenes [57]. Research indicates that the effective inhibition of ATR can lead to an increase in replication stress in malignancies driven by oncogenes, thus resulting in an increase in replication damage [34,57,58]. The overexpression of the Ras oncogene, which occurs in 90% of instances and promotes tumor cell survival, is one of the hallmarks of cancer cells [59,60,61]. The inhibition of ATR caused a rise in the level of replication stress, leading to cessation of tumor development. A significant mortality of tumor cells has been achieved through the combination of Ras activation and total ATR inhibition [38]. In order to exploit the effect of ATR suppression on cancer treatment, an ATR-Seckel mouse model has been used [30,62]. In this mouse model, a hypomorphic mutation of ATR, similar to the mutation that causes Seckel syndrome in humans, is present, resulting in a profound decrease in ATR activity. Impaired replication of DNA in embryogenesis and premature ageing due to stress accumulation caused by the deletion of p53, which was exacerbated after removal of p53, has been shown in the ATR-Seckel mice model [62]. Microcephaly and premature senescence were two phenotypic hallmarks of the ATR-Seckel mouse model. Overexpression of the Myc oncogene resulted in increased embryonic mortality in ATR-Seckel animals as well as worsened microcephaly and early senescence [36]. In this mouse model, however, the development of Myc-induced lymphomas or pancreatic tumors, which exhibit elevated levels of replication stress, was averted [36]. Chk1 inhibitors also inhibited tumor growth in these Myc-induced tumors, but no response was shown in K-RasG12V pancreatic adenocarcinomas, which lacked replication stress [36]. Similar research by the same group and Brown and colleagues investigated a genetic method to conditionally limit ATR expression in adult mice to 10% (hypomorphic suppression) of normal levels [33]. They discovered that in these animals, the growth rates of fibrosarcoma’s expressing H-rasG12V and acute myeloid leukemias (AMLs) driven by MLL-ENL and N-rasG12D [33] were dramatically reduced. Normal tissues, such as bone marrow and intestinal epithelium, were not harmed in these mice, showing the tumor selectivity potential of ATR inhibition.

## 3. Cytoplasmic Functions of ATR

Apart from ATR nuclear function regulated through the kinase domain, the cytoplasmic ATR was discovered to serve a significant antiapoptotic effect directly at the mitochondria, independent of nuclear ATR and kinase activity [63]. The nuclear kinase and cytoplasmic antiapoptotic functions of ATR are carried out independently by two prolyl isomeric forms of ATR, namely, *trans*- and *cis*-ATRS428P429 [63]. Both of these variants rely on a single-peptide bond orientation in ATR via a prolyl isomerization motif at S428P429 in humans. ATRIP is consistently missing in the cytoplasm, except during its production. In contrast to nuclear ATR, which is constantly in the trans form in a complex with ATRIP, cytoplasmic ATR, which lacks ATRIP, is mostly in the cis isomeric form after DNA damage [63]. Pin1 and PP2A balance the cis and trans cytoplasmic versions of ATR, with the former facilitating the conversion of *cis*- to *trans*-ATR by detecting the phosphorylated Serine 428–Proline 429 motif (pS428-P429) in ATR’s N-terminus [63,64]. Although Pin 1 activity promotes *trans*-ATR synthesis, DAPKI kinase inactivates it in response to DNA damage. In addition, DNA damage activates PP2A, which dephosphorylates ATR (pS428-P429), not only disabling the S428 phosphorylation-dependent Pin1 activity on ATR but also leading to *cis*-ATR formation ATR [64,65], as ATR is naturally stable in the *cis*-isomeric form of ATR [65]. Unlike the *trans*-ATR isoform, the *cis*-ATR includes an exposed BH3-like domain, which allows it to connect to the proapoptotic tBid protein at the mitochondria [63,64,65,66,67] (Figure 2). This binding prevents tBid from triggering the Bax–Bak polymerization pathway, which is necessary for the intrinsic apoptotic pathway. From here on, *cis*-ATR performs an antiapoptotic function, allowing the cell to survive long enough to repair its damaged DNA [64]. It is of importance to investigate whether *cis*-ATR might be used as a potential target for developing effective novel anticancer drugs.

## 4. ATR Regulation of Nucleus Mechanics

Nuclear mechanosensing and mechanical features of the nucleus influence genome integrity, nuclear architecture, gene expression, cell migration, and differentiation [68,69]. ATR is a transcription factor that responds to mechanical stress at the nuclear envelope and mediates the envelope-associated repair of aberrant topological DNA states. Alteration of nuclear flexibility and YAP delocalization is observed in ATR-defective cells. ATR-defective nuclei collapse when subjected to mechanical stress or interstitial migration, accumulating nuclear envelope ruptures and perinuclear cGAS, indicating loss of nuclear envelope integrity and an abnormal perinuclear chromatin state. Furthermore, during development and in the metastatic spread of circulating tumor cells, ATR-defective cells have a defect in neuronal migration. Furthermore, the mechanical coupling of the cytoskeleton to the nuclear envelope and the accompanying regulation of the envelope–chromosome association is ensured by ATR [70].

## 5. ATR Regulation of Autophagy

DNA damage has been linked to autophagy via ATR/Chk1/RhoB-mediated lysosomal recruitment of the TSC complex and subsequent mTORC1 suppression. UV light (UV) or the alkylating chemical methyl methane sulphonate (MMS) damage DNA, causing Chk1 to phosphorylate the small GTPase RhoB. Phosphorylation of RhoB boosts its interaction with TSC2 and sumoylation by PIAS1, both of which are essential for the RhoB/TSC complex to translocate to lysosomes. mTORC1 is thereby blocked, and autophagy is induced. RhoB knockout drastically reduces TSC complex lysosomal translocation and DNA damage-induced autophagy, both of which are dependent on ATR–Chk1-mediated RhoB phosphorylation and sumoylation [71]. By acting on the transcriptional factor GATA4, ATR has also been associated with transcription regulation [72], which is susceptible to degradation by the autophagy-related factor p62 (also known as sequestosome 1 (SQSTM1)). In fibroblasts, ATR has been reported to have a negative effect on p62 [73]. In addition, a reduction in GATA4 levels and changes in gene expression associated with senescence and inflammation are observed when cells are treated with an ATR inhibitor [73].

## 6. The *cis*-ATR Anti-Apoptotic Role Drives Oncogenesis in Dividing Cells

Cancer is characterized by deregulated cell proliferation, which develops when there is an imbalance in the normal cell cycle regulation to govern the pace and integrity of cell division and growth. Furthermore, because *cis*-ATR has antiapoptotic properties, we surmise that it may have an oncogenic role, whereas Pin1 may have tumor-suppressive properties in relation to ATR’s anti-apoptotic activity at the mitochondria. If *cis*-ATR is the dominant cytoplasmic form, it may inhibit mitochondrial apoptosis, allowing injured cells to survive and mutate even when DNA damage repair is insufficient, and the aberrant cells are intended to die by apoptosis. This evasion of apoptosis is a key feature of cancer cells, allowing them to accumulate the mutations that define genomic instability and, eventually, carcinogenesis. However, if Pin1 activity is elevated and *trans*-ATR is the dominant form of ATR in the cytoplasm, programmed death will occur in cells that are too badly damaged for effective DNA repair before mutations can be transmitted. Thus, lowering cytosolic *cis*-ATR prevents the accumulation of cells with DNA damage, which could be passed on to daughter cells and induce carcinogenesis.

The current knowledge is based mostly on data showing that Pin1 is overexpressed/activated in most malignancies and cancer stem cells, with correspondingly poor prognoses [72,74,75,76,77,78,79,80]. Pin1 also promotes the expression of multiple oncogenes while suppressing the expression of several tumor suppressor genes [81]. Pin1 overexpression or activation can be blocked genetically or chemically with juglone [82], all-trans retinoic acid [34] (ATRA), or KPT-6566 [83], and Pin1 inhibitors have been shown to reduce malignancies when tested [84,85,86,87,88,89,90]. However, chemically suppressing Pin1 presents numerous complications, particularly with retinoids (e.g., ATRA), the most used clinical inhibitor. Low medication bioavailability, clinical relapse, and retinoid resistance are examples of these [91,92,93,94,95]. Bioinformatic analyses of human tumors (Kaplan–Meier Plots) reported in the Human Protein Atlas (7932 cases) showed that low Pin1 RNA expression correlates with a lower survival profile for the majority (12 types) of cancer patients, whereas high expression correlates with a higher survival profile for three types of cancer. The association between survival profile and Pin1 expression in two other forms of cancer is unknown. Surprisingly, prostate and testis cancers are among the three minority groups; patients with low vs. high Pin1 RNA expression had a superior survival profile. These findings are also in line with the 5-year survival odds. However, only two cancer types, renal and pancreatic, are Pin1 expression prognostic: high Pin1 expression is associated with a better prognosis, according to the Human Protein Atlas. This is in contrast to a recent report on the prognostic value of Pin1 in cancer, which examined data from 20 published studies involving 2474 patients and concluded that Pin1 overexpression was significantly associated with advanced clinical stage of cancer, lymph node metastases, and poor prognosis, though no correlation with poor differentiation was found [96]. It is interesting to note that mutations in p53 are known to occur in more than 50% of cancer types, and Pin1 expression has been shown to support mutant p53-induced oncogenesis [72]. Moreover, Pin1 regulates the wild type p53 function in DDR and also isomerizes it for wild type p53 [97,98,99]. Therefore, the p53 status can have a bearing on the relationship between Pin1 expression and cancer, as there may be different effects for mutant or wild types of p53 in tumor cells [100]. It is unclear whether or how p53 status affects cancer prognosis in relation to Pin1 expression levels, which is of great interest to determine. To provide a better context, we believe that further consideration and investigation should be given to the wider role played by Pin1 and its regulatory partners in carcinogenesis [101].

Following UV-induced DNA damage, the phosphatase PP2A dephosphorylates ATR at Ser428. Following UV irradiation, PP2A was shown to be involved in the decrease of pATR (S428) in the cytoplasm [64]. Because Pin1 function needs the Ser428–Pro429 recognition site in ATR to be phosphorylated, PP2A inhibits Pin1 activity by depleting cytoplasmic pATR at S428. Another layer of regulatory complexity to the DDR process is associated with this PP2A monitoring for ATR phosphorylation at the Pin1 motif recognition. In the event of DNA damage, PP2A interacts with ATR to dephosphorylate its Pin 1 recognition motif to prevent further isomerization of cis to trans ATR in the cytoplasm, resulting in accumulation of cis ATR-H in the cytoplasm. It was found that the accumulation of cis ATRH in the cytoplasm and mitochondria, following DNA damage, is associated with PP2A activity. Mitochondrial translocation ATR-H is an antiapoptotic protein that interacts with tBid to inhibit apoptosis activation and to reduce apoptotic cell death caused by DNA damage [64].

## 7. ATR: A Therapeutic Target

### 7.1. Chemotherapeutic Effect by ATR–CHK1 Inhibition

The ability of ATR to limit oncogenesis in its early stages, by acting as a barrier to the proliferation of aberrant cells, occurs principally through p53 activation. The activation may lead to DNA repair and checkpoint arrest [102]. The fundamental theory is that ATR/CHK1 inhibitors, particularly in p53-deficient cells, increase tumor cell death by cytotoxic drugs or radiation by interrupting cell cycle checkpoints [103,104]. UCN-01 (7-hydroxystaurosporine) is the first CHK1 inhibitor with broad-spectrum activity against the protein kinase C family. In addition, UCN-01 lacks selectivity and has a long half-life, limiting its potential applications, since it binds to alpha acidic glycoprotein, which causes hyperglycemia, and has a long half-life [105,106]. XL844, an ATP-competitive inhibitor of CHK1, CHK2, VEGFR-2, and VEGFR-3, is quite effective. XL844 is designed to prevent CDC25A degradation, bypass S-Phase checkpoints, and increase DNA damage when used with gemcitabine. In the case of xenografts and in vitro, XL844 increases gemcitabine activity. The clinical study of XL844NCT00479175 and NCT00234481 was called off for reasons that are not yet known [107,108]. The roles of natural medicines that target ATR and CHK1 have been comprehensively studied in preclinical and clinical settings. The clinical success of olaparib, an inhibitor of the PARP family of ssDNA repair proteins and the first DDR inhibitor to be approved by the FDA, has led to attempts to examine other DDR targeting drugs. In this intriguing strategy, little progress has been achieved in screening various natural compounds for ATR–CHK1 inhibition. Natural ATR–CHK1 activators and inhibitors have been shown in preclinical research to be clinically useful for targeted cancer therapy. ATR–CHK1 suppression causes severe G2 and S-M checkpoint abnormalities, causing cells to enter mitosis early, resulting in cell death. In addition, ATR–CHK1 signaling is necessary for precise chromosomal segregation during mitotic cell division, so life cannot exist in the absence of ATR–CHK1 signaling. This makes this pathway a viable target for cancer therapeutics, especially in cancers where this pathway is particularly vulnerable [7,102,109,110,111,112]. Potential ATR inhibitors and their respective concentrations among the various cell lines used are summarized in Table 1.

### 7.2. ATR Kinase Inhibitors

ATR belongs to the PI3K-related kinase (PIKK) family of enzymes, along with ATM, mTOR, DNA-PKcs, SMG1, and PI3K (Phosphatidyl Inositol 3 Kinase). Caffeine was the first ATR inhibitor to be identified that disrupted DNA damage induced cell cycle arrest and sensitized cells with DNA damage [117,128]. However, since the concentrations that prevent ATR and ATM are toxic, this medicine cannot be used in clinical trials, as it is also preventing ATM and PI3K family members.

The identification of potent and selective inhibitors for this family of PIKs was delayed in the development of inhibitors against more traditional serine, threonine, and tyrosine kinase due to their atypical nature. Schisandrin B was found to be a weak ATR inhibitor with more specificity than caffeine in early screening studies. In recent years, NU6027, NVP-46BEZ235, Torin 2, and ETP-46464 have been reported to be more potent ATR inhibitors that have been shown to sensitize cancer cells to a variety of genotoxic chemotherapies but have also inhibited other kinases such as CDK2, PIK3, mTOR, and ATM [102].

The first potent and selective inhibitor, VE821, was reported by Vertex Pharmaceuticals in 2011. In comparison to ATM, PI3K, DNA-PK, and mTOR [129], VE-821 has >100-fold greater selectivity for ATR, and its analog VE-822 (VX-970) has been further enhanced with higher solubility, potency, selectivity, and pharmacodynamic characteristics. Preclinical studies showed that these agents potently sensitize several cancer cells lines to cisplatin, ionizing radiation, gemcitabine, PARP inhibitors, and topoisomerase I poisons etoposide and oxaliplatin in vitro [130,131,132,133,134,135]. The results of the xenograft studies are remarkable, namely, VE821 and VX970. VE-821 and VX-970 synergized with radiation and gemcitabine in pancreatic cancer xenograft models (Abdel-Fatah et al., 2015) and with irinotecan in a colorectal cancer model [131]. Like this, VX-970 made cisplatin more sensitive to six out of seven patient-derived primary lung tumor xenograft models, including tumors that had previously been resistant to the drug. In particular, the general toxicities of genotoxic treatment were not increased by adding VE821 or VX970 [133].

Very selective, potent ATR inhibitors are also in development by AstraZeneca. In 2103, AZ20 was first reported [43]. The analog of AZ20, AZD6738, was described in 2013 as an orally accessible drug with improved solubility and pharmacodynamic qualities [136]. Compared to other types of PIKK, AZD6738’s selectivity is excellent for ATR [137]. In vitro, both compounds exhibit single-agent efficacy in p53- and ATM-deficient tumor models, as well as sensitization to or synergy with gemcitabine, cisplatin, ionizing radiation, and PARP inhibition [46,47,136,137,138]. AZD6738 demonstrated monotherapy activity in primary CLL patient-derived xenografts with 11q deletion (ATM-deficient) and 17p deletion (p53-deficient), as well as xenografts of ATM- and p53-deficient mantle cell lymphoma cell lines [139]. AZD6738 cooperates with carboplatin, bendamustine, and cyclophosphamide in an ATM-deficient diffuse large B-cell lymphoma model [137] and a primary triple negative breast cancer explant when neither AZD6738 nor the PARP inhibitor displayed anticancer activity as single therapies [137].

### 7.3. ATR Kinase Inhibitors in Clinical Trials

In 2009, the first report on ATR-selective small-molecule inhibitors was published [118]. Schisandrin B, a naturally occurring dibenzo cyclooctadiene lignan found in the medicinal herb *Schisandra chinensis*, was found to be a selective inhibitor of ATR [118]. Schisandrin B inhibited UV-induced intra-S-phase and G2/M cell cycle checkpoints, and the cytotoxicity in human lung cancer cells was increased upon UV radiation. The inhibitory potency against ATR, on the other hand, was modest and required the use of large drug concentrations (30 M for cellular tests).

A more potent ATR inhibitor, NU6027, was reported in 2011 and was demonstrated to sensitize several breast and ovarian cancer cell lines to IR (insulin resistance) and several chemotherapeutic agents. However, this drug was originally created as a CDK2 inhibitor and is not ATR selective.

Toledo et al. also published the findings of a cell-based chemical library screening technique for the identification of effective ATR inhibitors in 2011 [34]. One of the compounds identified to possess significant inhibitory activity against ATR kinase was NVP-BEZ235, a drug originally introduced as a highly potent dual inhibitor of PI3K and MTOR with considerable in vivo anti-tumor activity [140]. NVP-BEZ235 has been demonstrated to be markedly radiosensitive to Ras-overexpressing tumors [141]. However, given that it also suppresses ATR (and to a lesser extent ATM and DNA-PKcs), it appears that DDR kinase inhibition, rather than PI3K or MTOR, led to the observed results. According to Gilad et al.’s research, ATR depletion is highly lethal in cells that overexpress oncogenic genes. Ras concurs with this viewpoint [38]. Another example of compounds with potent ATR inhibitory action but without selectivity are ETP-46464 and Torin 2 [34,142].

Vertex Pharmaceuticals discovered the first class of effective and selective ATR kinase inhibitors during a high-throughput screening strategy [133]. VE-821 was found to be a powerful inhibitor of ATP-like effects on ATR, but it had little to no interaction with other PIKKs such as ATM, DNA, and MTOR [44]. In the colorectal cancer cell line HCT116, VE-821 reduced phosphorylation of the ATR downstream target CHK1 at Ser345 and showed excellent constructive collaboration with genotoxic drugs from several classes. Chemosensitization was particularly evident with DNA cross-linking agents such as cisplatin, and it was further improved by p53 knockdown in ATM-deficient cells or in conjunction with the specific ATM inhibitor KU-55933. Importantly, VE821 showed little evidence of cytotoxicity in normal cells and caused only a reversible halt to growth without any significant induction of cell death [44]. These findings are also based on a study presented in the same year, which showed that gene inhibition of ATR expression selectively increases sensitivity to cisplatin in colon cancer cells with inactive p53 [143].

These findings support the idea that cancer cells lacking the G1 checkpoint are more vulnerable to ATR kinase inhibition, especially when combined with genotoxic therapies. VE-821 has since been used in several studies and has consistently been shown to sensitize a variety of cancer cell lines to IR and chemotherapy [51,52,131]. VE-821 was shown to increase the cytotoxicity induced by IR in a group of 12 human cancer cells, according to Pires et al. VE-821 radiosensitized cancer cells in the presence of severe hypoxia and at a variety of oxygen concentrations [51]. It is important to note that hypoxic tumor cells are more resistant to radiotherapy [144,145] and thus represent a major obstacle to its effectiveness. The efficacy of such a strategy in vivo remains to be determined.

In vitro radiosensitization of pancreatic cancer cell lines was demonstrated by VE-822, a counterpart of VE-821 with enhanced potency and selectivity against ATR, better solubility, and favorable pharmacokinetic features. VA-822 also significantly radiosensitized human pancreatic cancer xenograft models and significantly prolonged the growth retardation induced by IR in combination with gemcitabine. It is of great significance that VE-822 was well-tolerated in mice and did not increase toxicity in normal cells and tissues [130]. The first selective ATR inhibitor to be clinically developed was VX-822, which is now referred to as VX-970. VX-970 (VE-822) was demonstrated to significantly sensitize a panel of non-small cell lung cancer (NSCLC) cell lines (but not normal cells) to a variety of DNA damaging drugs (i.e., cisplatin/oxaliplatin/gemcitabine/etoposide) and SN38 (IRINOTECAN active metabolite). The sensitizing impact of VX-970 was most visible when cisplatin and gemcitabine were combined, sensitizing more than 75% of the 35 examined cell lines. The observed chemosensitivity was greater in cells having p53 depletion, as has been reported before, compared to cells with intact p53 activity. VX-970 substantially increased cisplatin responses in patient-derived lung tumor xenograft models (in six of the seven types) [134]. VX-970 has the potential to boost the effectiveness of DNA damage therapy in lung cancer patients, according to the research. Phase I clinical studies are currently being conducted to evaluate VX-970’s safety, tolerability, and pharmacokinetics in relation to cytotoxic chemotherapy (Source: ClinicalTrials.gov) (NCT02157792).

AstraZeneca’s AZD6738 is a second ATR inhibitor now in clinical development. AZD6738 is an analogue of AZ20, a strong and selective ATR inhibitor that has been demonstrated to have significant single agent activity in vivo at well-tolerated doses in MRE 11A-deficient LVO xenografts [42,146]. AZD6738 possesses significantly improved solubility, bioavailability, and pharmacokinetic properties compared to AZ20 and is suitable for oral dosing [140]. In vitro, it suppresses the phosphorylation of the ATR downstream target CHK1 while boosting the phosphorylation of the DNA DSB marker γH2AX. The combination of this compound with carboplatin or IR has shown significantly increased antitumor growth inhibitory activity in in vivo studies. AZD6738 also showed single-agent anti-tumor efficacy in ATM-deficient but not ATM-proficient xenograft mice [136,147]. This anti-tumor effect was linked to a sustained rise in γH2AX staining in tumor tissue but only a transient increase in normal tissues such as bone marrow or the gut. This shows that a positive therapeutic index might be obtained, which is optimistic for the future clinical development of this drug. A Phase I clinical study is currently being conducted to evaluate the efficacy and safety of AZD6738 administered on its own and in conjunction with radiotherapy for the treatment of patients with solid tumors (NCT02223923).

## 8. ATR Isomerization Potential Therapeutic Target

The most commonly used ATR inhibitors in cancer clinical studies are specific inhibitors of the enzyme ATR kinase, which plays a key role in the DNA damage checkpoint function of ATR in the nucleus. These inhibitors have no effect on *cis*-ATR’s antiapoptotic activity, since the new inhibitor of *cis*-ATR at mitochondria, which is independent from ATR kinase activation [63], is not affected. Such a protein target that is novel and effective in the treatment of cancer could be *cis*-ATR (ATR-H), potentially. *Cis*-ATR is not, as a matter of principle, mutagenic, but it allows cancer cells to evade apoptotic activation, which is particularly important for carcinogenesis. Cancerous cells may be resistant to death because they contain proportionally more cytoplasmic *cis*-ATR than healthy cells, especially when exposed to chemo- or radiotherapy, or because they have less Pin1 or less Ser428 phosphorylation in ATR [148]. In support, reduced levels of pSer428 ATR in the cytoplasm of advanced epithelial ovarian cancer cells are correlated with poor prognosis [149]. As a result, using irradiation or chemotherapy to target *cis*-ATR as an adjuvant treatment for cancer should preferentially kill *cis*-ATR-dependent cancer cells while having little influence on the normal activities of nuclear *trans*-ATR in cells. To guarantee cellular survival and normality, ATR is a crucial protein [31] that consists of cis and trans isomers that are normally active but exist in a fragile balance. We propose using *cis*-ATR as a novel, potential cancer treatment target by taking advantage of the natural balance that exists in normal human cells between *cis*-ATR and *trans*-ATR isoforms. In addition, *cis*-ATR could be a diagnostic marker for prognosis and efficacy in the treatment of cancer.

PP2A plays a fundamental role in the regulation of DNA damage response (DDR) through the dephosphorylation of DDR proteins to alter the phosphorylation state of proteins essential for genome stability [150,151,152,153,154,155]. A variety of PP2A substrates are present in DDR, including ATR, ATR-ATM, DNA-PK (DNA-PK), Chk1 (Chk2), and p53, among others [156]. When a DNA damage checkpoint signal is triggered, all of these regulations are carried out in the nucleus. For instance, when PP2A is activated, it attenuates ATR and ATR-dependent DDR [153]. PP2A inhibition or knockdown increases y-H2AX, an essential DNA damage marker of DNA strand breaks [157,158,159]. Dephosphorylation of ATM at S1981 is achieved by the action of PP2A or Wip1, which results in the suppression of ATM activity [156,160]. PP2A is also involved in regulating cell death [161,162,163]. The presence of PP2A makes it an appealing target for the sensitization of tumor cells to radiation. However, there is currently no FDA-approved inhibitor for PP2A. Known chemical inhibitors such as okadaic acid, cantharidine, etc., are toxic and have little clinical benefit [164]. A new drug, LB-100, is a small molecule inhibitor of PP2A. LB-100 was well tolerated in adult patients with progressing solid tumors in a completed Phase 1 study [165]. LB-100 was found to be an effective radiation sensitizing agent in a variety of preclinical trials [68], including pancreatic cancer [166], nasopharyngioma [166], and glioblastoma [167]. However, there was no study to examine the effect of inhibiting PP2A on meningioma, nor have any intracranial tumor models been investigated for LB-100’s radiosensitivity. To remedy this, in vitro studies were conducted using three immortalized meningioma cell lines (IOMM-Lee cell line, GAR cell line, CH-157 cell line) as well as IOMM-Lee cells for the purpose of constructing an intracranial skull base xenograft model [168].

## 9. Prospects

Anticancer medicines such as DNA-damaging methylating/alkylating compounds and topoisomerase inhibitors that activate ATR through phytochemicals may sensitize cancer cells. The clinical use of ATR/CHK1 inhibitors in the treatment of targeted cancer with or without combination therapy with chemotherapy and radiotherapy is supported by extensive preclinical data. In the majority of combinations, though, there is a lack of significant results from clinically conducted studies. As a single agent, the clinical efficacy of inhibitors of ATR and CHK1 is not known. Identifying the proper patient group to target and developing more highly selective ATR/CHK1 inhibitors is one of the keys to enhancing the efficacy of ATR/CHK1 inhibitors. There is a need to identify new synthetic lethal interactions between ATR and CHK1 inhibitors and other inhibitors targeting DDR pathways. The fast advancement of high-throughput synthetic lethality screening and next-generation sequencing could significantly increase the capacity to validate unknown targets. However, only a tiny fraction of possible synthetic lethality cases has been established. Additionally, it is essential for clinical implementation that new generations of pharmacological inhibitors with improved bioavailability and substrate specificity are developed. Lastly, optimized schedules are also to be used for the effective use of inhibitors against ATRs and CHK1 in clinic studies with a view to identifying a biomarker that is appropriate for these agents. The role of ATR and CHK1 inhibitors in the treatment of cancer will be revealed through results from current and planned clinical studies.

In addition, the phosphorylation status of pATR at S428 in cytopenia can change the influence of ATR on regulatory changes for the mitochondria cell death pathway. The modulation of this ATR phosphorylation should have limited effect on the DNA damage signal pathways in the nucleus because PP2A regulation is confined to the cytoplasm [64] and therefore has a low impact on pATR S428’s status. This implies that PP2A may be a possible target for regulating the activity of ATRs against apoptotic cells. If PP2A is inhibited, a significant increase in phosphorylation of ATR at Ser428 results in the formation of trans ATR with decreased cis ATR throughout the cytoplasm. Consequently, UV damage to DNA results in reduced protection for cells from apoptotic death. This might be a possible pharmaceutical target for controlling the mitochondria linked with ATR through phosphatase PP2A, which could lead to novel treatment techniques for apoptotic cell death and cancer.

## 10. Conclusions

The cellular functions of ATR are supported by two major activities: its kinase activity, and kinase-independent antiapoptotic activity directly at the mitochondria. ATR functions as a basal kinase and is an important part of repairing DNA, regulating the cell cycle, and apoptosis. It has a wide variety of therapeutic functions. Although research into the mechanisms of ATR signaling is still ongoing, we now have much better knowledge of ATR’s function in replication stress and other components of the DNA damage response (DDR). In conclusion, ATR kinase has emerged as a promising target for cancer therapy, and the area of ATR inhibition (ATRi) is quickly expanding, with multiple early-phase clinical studies now underway. This review confirms that in cancer therapy, the of ATR–CHK1 pathway has been targeted with a variety of compounds that originate from naturally occurring substances. Moreover, these compounds have demonstrated synergistic potential as chemotherapeutic agents when used in combination with other anticancer drugs. Alternatively, directly antagonizing the antiapoptotic *cis*- or mitochondrial ATR by inhibiting PP2A may reduce the apoptotic threshold of cancer cells, thereby promoting cancer cell death. Selectively moderating the levels of ATR kinase (*trans*-ATR) and antiapoptotic *cis*-ATR could be a potential strategy for enhancing cancer therapies.

## Figures and Tables

**Figure 1 ijms-24-11684-f001:**
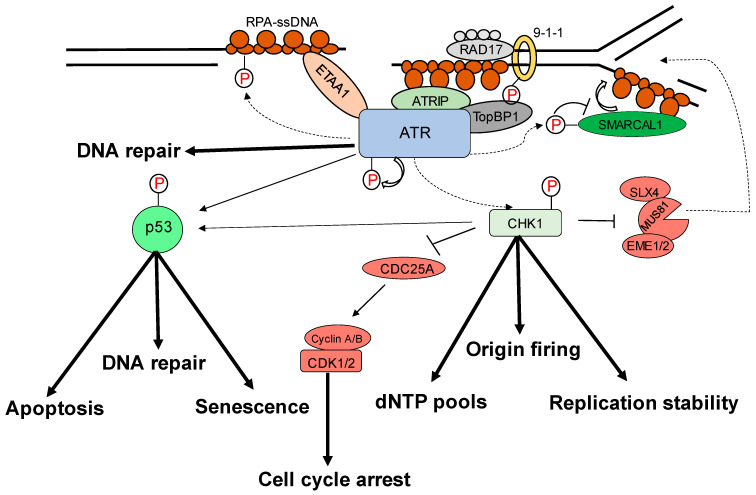
Replication stress-induced ATR–CHK1 activation. ATR is activated by RPA-coated single-stranded DNA (ssDNA) that forms at a stalled replication fork or resected DNA double-strand break (DSB), particularly at the ssDNA/dsDNA confluence. ATR-interacting protein (ATRIP) recruitment results in ATR and RPA-ssDNA complex recognition. It then integrates Rad9–Rad1–hus1 (9-1-1) and DNA topoisomerase 2-binding protein 1 (TOPBP1), activating ATR. ATR phosphorylates checkpoint kinase 1 (CHK1) via the adaptor protein claspin. CHK1 activation can help to prevent genomic instability. The processes promote or prevent the initiation of DNA replication (origin firing), ensure a sufficient supply of deoxynucleotides (dNTPs), stabilize the replication fork, and repair DNA. The signal is sent by transducer proteins to effector proteins such as p53, which is phosphorylated by ATR and CHK1. Cell-cycle arrest, DNA repair, apoptosis, and senescence are all mediated by p53.

**Figure 2 ijms-24-11684-f002:**
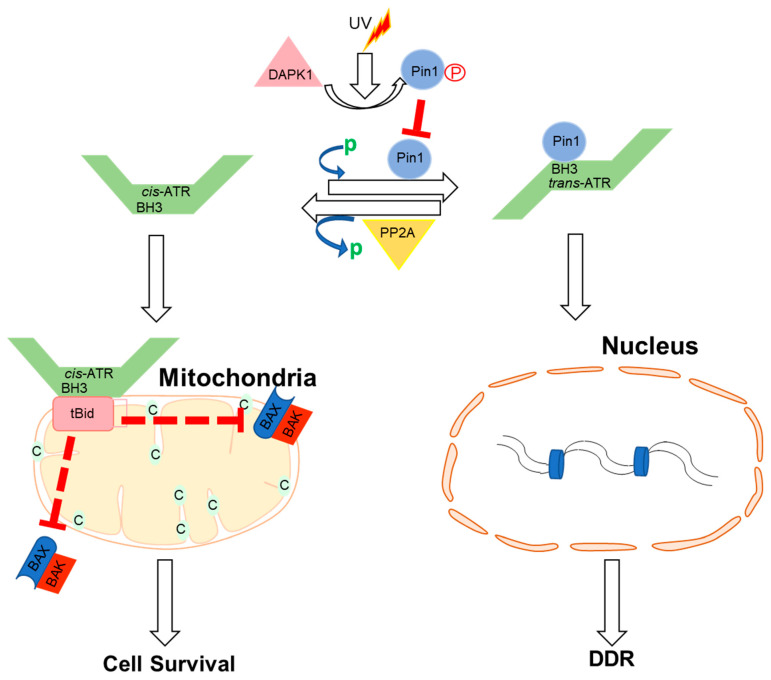
Proposed mechanisms by which ATR plays a direct anti-apoptotic function at the mitochondria. UV irradiation inhibits Pin1’s isomerization of ATR in the cytoplasm. *Cis*-ATR (ATR-H) then accumulates at the outer mitochondrial membrane, where it binds to and sequesters t-Bid. Because Bax and Bak cannot polymerize in the absence of tBid, *cis*-ATR suppresses cytochrome c release and apoptosis. In the nucleus, *trans*-ATR is the dominant isomer, interacting with ATRIP, RPA, and chromatin in the DNA damage repair (DDR) response. Protein phosphatases (PP2A) can dephosphorylate the Pin1 recognition motif and induce *cis*-ATR synthesis.

**Table 1 ijms-24-11684-t001:** Potential ATR inhibitors.

Drug	Mode of Action	IC_50_	Cell Type	References
Rocaglamide-A (Roc-A)	Roc-A induces the rapid degradation of Cdc25A by activation of the ATM/ATR–Chk1/Chk2 checkpoint pathway	50–100 nM	Acute T cell leukemia lines Jurkat-16, CEM, Molt-4, and DND-41; the T lymphoma cell line Hut-78; the acute myeloid leukemia cell line HL-60; the Hodgkin lymphoma cell line L1236; the hepatocarcinoma cancer cell lines HepG2 and Huh7; the colorectal cancer cell lines HT-29 and HCT116; the prostate cancer cell line PC3; and the breast cancer cell line MCF-7	[113,114]
Harmine	Harmine, a natural compound, negatively regulates HR but not NHEJ by interfering with Rad51 recruitment, resulting in severe cytotoxicity in hepatoma cells	20 μM	Hep3B and HuH7 cell	[115]
Caffeine	Disrupted the G2-M and hastened the transition to mitosis, culminating in apoptosis in A549 cells	1.1 mM	BCR/ABL leukemia cells	[116,117]
Schisandrin B (Sch-B)	inhibit ATR in A549 adenocarcinoma cells	7.25 μM	A549 cells	[118]
Kaempferol	promoted DNA damage in HL-60 cells	75 μM	HL-60 cells	[119]
Curcumin	Curcumin-induced DNA demethylation of human gastric cancer cells (hGCCs) was mediated by the damaged DNA repair-p53-p21/GADD45A-cyclin/CDK-Rb/E2F-DNMT1 axis	20 μM	MGC-803 gastric cancer cells	[120]
	Curcumin suppresses three DDR pathways by inhibiting histone acetyltransferases and ATR. Concordantly, curcumin sensitizes cancer cells to PARP inhibitors by enhancing apoptosis and mitotic catastrophe via inhibition of both the DNA damage checkpoint and DSB repair	493 nM	HeLa (cervical cancer), U2OS (osteosarcoma), HT1080 (fibrosarcoma), HCT116 (colorectal cancer) cells and H1299 (non-small-cell lung carcinoma)	[121]
Resveratrol	Resveratrol causes Cdc2-tyr15 phosphorylation via the ATM/ATR–Chk1/2–Cdc25C pathway as a central mechanism for S phase arrest	50 μM	human ovarian carcinoma Ovcar-3 cells	[122]
	Impedes the Rad51, BRCA1 (breast cancer 1), and BRCA2 expression involved in ATR–CHK1 activation	150 μmol/L	MCF-7 cell	[123]
Apple peel flavonoid fraction (AF4)	The apple peel flavonoid fraction (AF4) protects BEAS-2B cells against different carcinogens, including nicotine-derived nitrosamine ketones, via decreasing ATR–CHK1 signaling	50 μg/mL	Normal human bronchial epithelial cells	[124]
Triptolide	Triptolide from *Tripterygium wilfordii* induces DNA damage in A375.S2 melanoma cells by ATR–CHK1 inhibition	15–30 nM	The human malignant melanoma cell line (A375.S2)	[125]
Mangiferin	Mangiferin induces cell cycle arrest at the G2/M phase through the ATR–Chk1 pathway in HL-60 leukemia cells	160 μM	HL-60 leukemia cells	[126]
Protoapigenone	The function of WYCs revealed that they have a potential ability to inhibit DDR, particularly on activation of Chk1 and Fanconi anemia group D2 protein (FANCD2), but not Chk2. In this way, WYCs inhibited ATR-mediated DNA damage checkpoint and repair	8 μmol/L	MDA-MB-231 (breast adenocarcinoma) and A549 (lung adenocarcinoma	[127]

## Data Availability

Not applicable.

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
