# Peer review of "Novel Cellular Functions of ATR for Therapeutic Targeting: Embryogenesis to Tumorigenesis"

_ijms, 2023, doi:10.3390/ijms241411684_

Round 1

Reviewer 1 Report

The manuscript "Novel Cellular Functions of ATR for Therapeutic Targeting: Embryogenesis to Tumorigenesis". The authors give a comprehensive review of the cellular function of the DNA repair signaling protein ATR. The give both cytoplasmic and nuclear functions of ATR. They also discuss potential therapeutic uses for ATR inhibitors in cancer.

The manuscript is clear written and of should be of benefit to the field.

Reviewer 2 Report

Inhibitors of ATR and other DNA damage response proteins have long been established as potential anticancer agents for cancer therapy. This paper provides a quick review of ATR cellular functions, describes the progress made in using inhibitors of ATR signaling pathways for cancer therapy, and discusses future therapeutic approaches for the clinic. I believe this paper will be of interest to cancer cell biologists. I only have a few specific comments to make:

1. There is no mentioning the functions of the ATR-Chk1 signaling axis in mitosis, which ensure the fidelity of chromosome segregation (e.g., doi: 10.1126/science.aan6490 and DOI:10.1083/jcb.201104023). I believe this is an important function of ATR and Chk1 that needs to be discussed and properly referenced.

2. Line 40: “our oncogene paper”: this is redundant, please delete.

3. Line 41: Please change to “…the RPA binding protein Ewing's tumour-associated antigen 1 (ETAA1)”, for clarity.

4. Line 52: “ATRChk1”, please change to: “ATR-Chk1”

5. Line 72: “embryonically fatal”, please change to “embryonically lethal”.

6. Lines 271-272: “The ability to limit oncogenesis in its early stages by acting as a barrier to the proliferation of aberrant cells, principally through p53 activation”. A verb is apparently missing, please rewrite.

7. Line 286: Please change to: “…Olaparib, an inhibitor of the PARP family of ssDNA repair proteins…”, for clarity.

8. Lines 450-452: “A new drug LB-100 and it's a new type of small molecule that inhibit PP2A. LB-100 was well tolerated in adult patients with progressing solid tumors in a completed Phase 1 research”. I believe something is missing, please rewrite.

9. Lines 456-457: “As a result, we decided to test LB100's 456 radiosensitizing capability in AAM in a pre-clinical context”. I believe this sentence is redundant, please delete.

10. Line 464: “ATR/CH1”, please change to ATR/CHK1.

The quality of English language is generally fine. Please also see my comments #6 and #8 above.
